# Non-Destructive Detection of Real Defects in Polymer Composites by Ultrasonic Testing and Recurrence Analysis

**DOI:** 10.3390/ma15207335

**Published:** 2022-10-20

**Authors:** Krzysztof Ciecieląg, Krzysztof Kęcik, Agnieszka Skoczylas, Jakub Matuszak, Izabela Korzec, Radosław Zaleski

**Affiliations:** 1Department of Production Engineering, Faculty of Mechanical Engineering, Lublin University of Technology, 36 Nadbystrzycka, 20-618 Lublin, Poland; 2Department of Applied Mechanics, Faculty of Mechanical Engineering, Lublin University of Technology, 36 Nadbystrzycka, 20-618 Lublin, Poland; 3Department of Materials Physics, Institute of Physics, Faculty of Mathematics, Physics and Computer Science, Maria Curie-Sklodowska University, Marii Curie-Sklodowskiej Sq. 1, 20-031 Lublin, Poland

**Keywords:** defect detection, non-destructive testing, recurrence plots, drilling, C-scan, polymer composites

## Abstract

This paper presents results of ultrasonic non-destructive testing of carbon fibre-reinforced plastics (CFRPs) and glass-fibre reinforced plastics (GFRPs). First, ultrasonic C-scan analysis was used to detect real defects inside the composite materials. Next, the composite materials were subjected to drilling in the area of defect formation, and measured forces were used to analyse the drilling process using recurrence methods. Results have confirmed that recurrence methods can be used to detect defects formed inside a composite material during machining.

## 1. Introduction

### 1.1. Non-Destructive Testing

Construction materials such as steel, aluminum alloys, titanium alloys, magnesium alloys and polymer composites are prone to develop real internal defects. Composites consist of reinforcement and resin as main components of this material. Defects such as pores, bubbles, inclusions and cracks in the reinforcement or resin are formed at the manufacturing stage. These defects are invisible on the surface of the material [1,2].

In order to find out about defects in materials, research on the structure of the material is carried out. Destructive and non-destructive tests are carried out to prevent negative effects of defects on the operation of machine components [3,4]. In destructive testing, a selected batch of elements is destroyed in order to identify the location of defects and qualify the usability of these elements. This type of testing involves a loss of several percent of produced items. The second type of material analysis is non-destructive testing. During non-destructive testing, the analyzed elements can be used without damage for further processes. Comparing the two types of tests, it was found that in addition to the criterion of sample destruction during destructive tests, the measurements are direct, they focus on quantitative measurement, it is not possible to repeat the test on the same specimen, which is destroyed and the preparation of the test sample and the test itself is expensive. Non-destructive testing is characterized by the fact that the measurements are indirect, focus on the qualitative measurement, the test can be repeated on the same sample, testing is carried out on a real part, many properties of the sample can be measured, measurements can be repeated multiple times, and the test is relatively quick [5]. Therefore, polymer composites are preferably subjected to non-destructive tests, in which tested elements are intended for further operation. Non-destructive testing (NDT) can be divided into two groups: contact methods and non-contact methods [6]. Contact methods include magnetic, electromagnetic, penetrant and traditional ultrasonic testing. In turn, testing methods such as shearography, radiography, thermography and visual inspection are non-contact methods [6]. This is the method by which the element is magnetized. Defects in the structure cause a magnetic current to penetrate the material [7]. The electromagnetic method uses magnetism and electricity to detect defects and changes in the material structure. Changes in the electric current and magnetic field are the basis for the assessment of the defect [6]. The disadvantage of magnetic methods is the detection of defects greater than 1 mm [8]. These are good methods for assessing the depth of defects due to the possibility of penetration of a few mm into the material [7]. Ultrasonic methods use high frequency sound waves. Defect detection consists in examining the wave reflected from the defect inside the material. The advantage of this type of test is that it is completely non-destructive. Ultrasonic testing allows the detection of defects over 0.5 mm [8]. Preparation of the sample for testing does not require costly preparation, and the test can only take place from one side. An additional advantage of this method is that the results are reproducible. However, the operation of measuring devices requires a skilled worker, and testing complex shapes is sometimes difficult [9,10,11]. Shearography testing is a laser optical method that is based on the degree of concentration of deformations around a specific defect. The advantage of this method is that it can be performed by less skilled workers and the method is not prone to noise. The disadvantage of this method is that it is mainly used to detect delamination [12]. Radiography testing using X-rays is a common method for detecting voids, cracks, inclusions and various types of fibre orientation. An important advantage of this method is the ability to detect defects at a level of 50 µm [13]. Thermographic testing using heat is based on the change in thermal conductivity at the location of the defects. The advantage of this method is the possibility of testing a large area, and for testing, access to only one side of the sample is sufficient. The disadvantage of this method is that it is not effective for use with thin materials [6]. Visual inspection is a very common method of non-destructive testing. The most important advantage of this method is its speed and relatively inexpensive equipment. The disadvantage of this method is that it can only be tested on a surface [14].

### 1.2. Ultrasonic Testing

Non-destructive methods can be used to test mechanical and physical properties of composites. They are also useful in evaluating the internal structure of composite materials, including the detection of defects inside the material. Numerous studies show that the use of non-destructive testing for composites makes it possible to determine their mechanical strength and stiffness, as well as to assess the effect of temperature on the shear strength of glass and carbon fibre composites based on epoxy resins [15]. Non-destructive testing also allows the assessment of the influence of moisture absorption on the mechanical properties of glass fibre-reinforced plastics [16]. Ultrasonic testing (UT) enables the determination of fibre distribution in composites and is a source of information about the amount and size of fibres making up the composite [17]. The use of ultrasound diagnostics for composite materials makes it possible to determine changes in their structure after thermal aging [18]. Non-destructive methods, including observation, are also successfully employed to assess damage evolution [19]. The tests that do not cause the destruction of the tested material are also used for assessing the connection of another structural element with the composite, the condition of resin after curing, and the relationship between resin and reinforcement [20,21]. The use of ultrasonic scanning and images of structures also makes it possible to determine indicators, on the basis of which the type and level of material degradation can be assessed [21].

A wide range of non-destructive testing applications, including ultrasonic testing, are successfully used to analyse defects in composites [6,18,21]. They made it possible to determine the location of structural cracks in the composite material and the location of extensions and cracks in the fibres. Moreover, they are used to determine the delamination front. Defects or discontinuities in the structure of a composite material, such as structural cracks, pores, inclusions and delamination, are effectively detectable by ultrasonic testing [22]. The study [23] reviews the damage modes and defects in aviation composites that have a significant negative impact on reducing the stiffness and strength of composite materials. Inspection, testing and non-destructive assessment of a composite material can be applied after the production of the composite as well as during its use and related maintenance works. Monitoring the condition of fibres and their defects largely contributes to ensuring the reliability of working components made of composite materials [24]. There have also been attempts to compare existing defects by means of non-destructive and numerical tests. Results demonstrate that numerical predictions by the finite element method are compatible with experimental measurements, including scanning electron microscopy (SEM) [24,25].

Tests, which result in obtaining C-scan images, belong to a group of non-destructive tests, namely ultrasonic tests. Results obtained from ultrasound tests include A-, B- and C-scan images [26,27]. The C-scan technique makes it possible to identify the areas of discontinuity, showing changes in the acoustic field. This method gives very accurate results through automated head movement (usually in two directions) together with a water acoustic coupling mechanism. The damping of the material in composites is also crucial, therefore the transmitting and receiving system of a flaw detector is very important. The C-scan method allows one to detect defects that are very difficult to spot using standard methods.

Numerous studies show images created by defectoscopic ultrasonic testing, including the Phased Array (PA) method. The heterogeneity of the material, which is commonly represented by bright (usually red and yellow) fields, serves as a barrier to the wave as it passes through the composites. Dark areas related to satisfactory absorption and refraction of waves (usually marked in dark blue) represent places where the structure of the material is homogeneous and free from defects [28,29]. The ultrasonic C-scan also enables the study of thin composite structures, and the quality of obtained images depends on the frequency of the transmitted waves [30]. This method is also used to monitor the structural condition of composites and steels in use.

The through-transmission method can also be used in ultrasonic testing [31], in which the transmitter and receiver are located on both sides of the tested material. The ultrasonic wave is sent from the transmitting head and passes through an element that is placed between the transmitting head and the receiving head. The location and measurement of defects is accomplished by the passage of ultrasonic waves, recording the transmission of the ultrasonic wave through the entire element, not just parts of the reflected waves. This method involves immersing the transmitter, receiver and sample in a liquid medium.

With the development of composites and the need to use them as a construction material, the techniques of non-destructive testing are also developed [32]. Many studies describing the application of the through-transmission method relate to composites used in industry [33]. Non-destructive tests are also used to test the elastic properties of composite materials [34,35]. The through-transmission method as well as the Phased Array (PA) method make it possible to detect defects inside composite materials [36]. Previous studies have demonstrated that defect orientation can have a significant impact on its detection. This, however, does not affect the fact that ultrasonic methods yield good results in terms of detection of true defects in composites [36]. Composite materials are also tested by thermovision and interferometric methods [37]. The thermovision method uses the observation of dynamic changes in the temperature distribution on the surface (thermal imaging camera) after heating the material with a strong heat pulse. Interferometric methods are based on the phenomenon of electromagnetic wave interference in order to visualize slight deformation of the material under the influence of mechanical load.

### 1.3. Non-Linear Method for Defect Detection

Non-linear data analysis is used in studies, in which there are many variables and thus results cannot be described using a simple linear function. One example of non-linear data analysis is the recurrence method [38]. In the recurrence method an important role is played by a recurrence plot [39]. According to the mathematical approach, a recurrence plot can be described by a formula:*R_i,j_* = *H*(*ε*−||*x_i_* − *x_j_*||), (1)
where, *i* and *j* are the numbers of the states in the phase space. State numbers take the values 1, …, N, where N stands for the number of states. In turn, *x_i_* and *x_j_* denote the vectors in the phase space, and the symbol *H* denotes the Heaviside step function. It is a function that takes a value of 1 for positive values and a value of 0 for negative values. The symbol ε is the threshold parameter and *||x||* is the norm, which is usually the Euclidean or maximum norm. Methodology of the determination of the threshold parameter is described in previous articles. Graphically, the recurrence plot is an array of points *i*, *j*, which is made up of white and dark dots. A square matrix with the dimensions N × N has symmetry with respect to the main diagonal, in which the numbers of states *i* and *j* are equal to each other. Descriptions of recurrence methods and delay parameter selection can be found in numerous research papers [40,41,42]. Recurrence quantifications complement recurrence plots. They were introduced to provide additional information about changes in tested elements or phenomena [42,43,44]. Descriptions of individual recurrence quantifications can be found in our previous research papers [45,46]. Recurrence methods are used not only in industry but also in medicine and finance. In medicine they are used to define the nature of heart work, contributing to an early classification of cardiovascular disfunctions [47]. In industry, specifically in machining, recurrence methods are used to study surface defects in polymer composites. In studies on the milling process, these methods proved effective in detecting the size and location of specific surface defects [48]. The use of recurrence methods yielded positive results in drilling tests for glass and carbon fibre-reinforced plastics [45]. Thanks to the use of non-linear analysis, it was possible to detect the position of an artificial defect in drilling.

### 1.4. Motivation and Aim

This work is a continuation and extension of our previous successful research. In our previous research, we initiated the innovative application of nonlinear research such as recurrence methods in the machining. It was found that, based on the analysis of cutting processes by recurrence methods, it was possible to create recurrence plots and select recurrence quantifications that enable the detection of artificial defects in polymer composites. The research initially focused on investigating the milling process, in which the position and size of artificial defects, constituting a few percent of the milled cross-section, were analysed. It was determined that among the many recurrence quantifications, the recurrence rate, determinism and laminarity were the most appropriate [46,48,49]. The drilling process was investigated in further studies. The studies also allowed for the selection of quantifications for investigating artificial defects [45]. It was shown that artificial defects formed during drilling had to be 12% (for GFRP) and 18% (for CFRP) of the cross-section of the drilled hole. The study also demonstrated that quantifications such as the averaged diagonal length (L), entropy (ENT) and recurrence time of the second type (T_2_) were the most universal indicators to be used in studies on drilling.

The aim of this study is to perform a non-linear analysis of the drilling process for glass and carbon fibre-reinforced plastics with real defects formed in the composite materials at the manufacturing stage (in our previous studies, the defects were artificial). This work employs recurrence methods to detect real defects, the presence of which was confirmed by non-destructive ultrasound tests. The study is also aimed at selecting recurrence quantifications that are indispensable to detect the location and size of a defect formed at the manufacturing stage. In this paper, for the first time in our works, research of real defects inside composite material by the recurrence method were carried out. This is important, because the real defect has a much more complicated shape. The location of the real defect was first confirmed by ultrasonic non-destructive testing.

## 2. Materials and Methods

### 2.1. Specimens, Tools and Equipment

Two types of polymer composites saturated with epoxy resin were tested. These were samples with the dimensions of 150 × 30 × 10 mm, consisting of 40 layers of prepregs arranged in a 0–90° system (alternating arrangement of fibres). One type of the composite material was carbon fibre-reinforced plastic (CFRP) with the trade name GR/EP 985-GF-3070, and the other was glass fibre-reinforced plastic (GFRP) with the trade name EGL/EP 3200-120. For both types of material, the resin accounted for 60% of the volume of the entire material. The samples were prepared in controlled environmental conditions with temperature (18–30 °C) and humidity (less than 60%) as well as adequate air purity (less than 10,000 solid particles per 1 m^3^). Next, the ready-made prepreg materials were placed in autoclaves, in which the temperature (177 °C) and pressure (0.3 MPa) were maintained for 2 h. After their removal from the autoclave, the samples were allowed to cool for 24 h and resin discharge was cut off. The surplus resin was removed by circumferential milling.

The composite materials were subjected to non-destructive ultrasonic testing by the 5 MHz through-transmission method using an automatic flaw detector, MIDAS CIJP 5/3. The through-transmission method consists of passing a beam of ultrasonic waves through the material placed between the ultrasonic transmitting head and the receiving head. In addition, the composite sample is soaked in a liquid medium. The ultrasonic heads are located inside the containers, to which pressurized water is supplied. Water flows out through the outlet nozzles, forming streams that reach the surface of the test stand. Ultrasonic pulses generated by the probe pass through the water jets to the surface of the tested element.

Based on damping levels, the system generates a C-scan image using shades of grey or a colour code. The generated damping map shows structures present inside the tested element as well as irregularities and defects in the material. By automating the process of recording the position of the heads, sound data are visualised with accuracy sufficient for the determination of changes in the level of acoustic properties. On the other hand, changes in the pulse amplitude (coupling variability) are reduced by automating the head travel. A schematic of the research methodology employed in the study using the through-transmission method and internal defect location is illustrated in Figure 1.

The non-destructive testing (NDT) was followed by a drilling operation. Drilling was carried out with a Kennametal’s drill (B041A10000CPG KC7325) with a diameter of d = 10 mm. It is a solid carbide drill coated with TiAlN-PVD to ensure the essential level of wear resistance. The drill with two blades had a helix angle of 30° and a point angle of 140°.

### 2.2. Machining and Measurements

The drilling process of the composite samples with detected internal real defects was conducted on the Avia-VMC 800 HS vertical machining centre. The plate had a thickness of 10 mm, which was the length of the hole made. The machining was conducted with a feed speed of *v_f_* = 287 mm/min and a cutting speed of *v_c_* = 60 m/min. A drilling feed force measurement system was installed in the working area of the vertical machining centre. The system consisted of a 3D dynamometer from Kistler (type 9257B), Kistler’s charge amplifier (type 5070), Dynoware data acquisition card (type 5697A), and Dynoware software (type 2825A). This equipment was used for measuring and processing signals in the form of feed force. The composite samples were bolted to the 3D Kistler dynamometer. An experimental setup for conducting tests by the through-transmission method is shown in Figure 2, while the machining process is shown in Figure 3.

Measured feed forces were then used for analysis using recurrence methods. The force tests were performed with an accuracy of 0.00001 N. The drilling forces were recorded every 0.0001 s. Results of those analyses were used to create recurrence plots and recurrence quantification diagrams that are extremely useful for determining the location of actual defects inside composite materials.

## 3. Results and Discussion

### 3.1. C-Scan Defect Detection

The first stage of the study was to detect the location of a real internal defect inside the composite materials. Using the automatic flaw detector MIDAS CIJP 5/3 and the through-transmission method, it was possible to obtain C-scan images for the two tested types of composite materials. Obtained results are presented in Figure 4.

The red fields in Figure 4 indicate the location of real internal defects (not introduced artificial internal defects as in our previous paper [45]) which are visible as disturbances during scanning. For the CFRP composite sample, we detected a real defect of 1.3 mm with the through-transmission method. On the other hand, the real defect detected in the GFRP composite sample was 1.1 mm in size.

### 3.2. Recurrence Analysis

#### 3.2.1. Recurrence Quantification Analysis of CFRP Drilling Signals

Drilling tests of GFRP and CFRP samples were conducted using the experimental system shown in Figure 3. The composite samples were drilled at two places: at the exact location of the defect detected by the C-scan method (which is marked as “defect”) and away from the location of the defect (marked as “no defect”).

Figure 5 shows drilled force signals obtained from CFRP drilling tests. It shows the selected section from the total drilling process. The section means the data points include an account of the entire defect and its immediate vicinity. The blue line marks the signal obtained from no-defect drilling, while the red line indicates the signal in defect drilling. Both signals show high frequency oscillations.

Both drilling forces oscillate between 450–600 N. A slight difference can be observed between 0–1500 data points, which indicates some dynamical changes in the drilling process.

A detailed analysis of the drilled forces was made by means of recurrence analysis. The RP method is employed to measure recurrences in the phase space of a dynamical system. Periodically oscillating systems have recurrence plots with diagonally-oriented recurrent structures (diagonal lines). However, abrupt changes in the system dynamics and extreme events cause white areas (no recurrence points) or bands in the RP structure. Therefore, this effect can be used to detect defects in the analysed signal.

Recurrence plots were obtained from the normalized drilled time series without defect (Figure 6a) and with defect (Figure 6b). The plots were created by using proper values of the embedding parameters: an embedding dimension of *m* = 5 (False Nearest Neighbour) and a delay of *d* = 2 (Average Mutual Information) after the normalization of the time series to zero mean and a standard deviation of one. As can be seen in Figure 6b, there is a place at some point of time (close to 500–1000) that may indicate abrupt changes in the system dynamics.

This place is probably responsible for the defect location. Based on this place we can estimate that the approximate depth of the defect equals 0.4 mm. This effect is not observed in the RP presented in Figure 6a.

Recurrence structures can also be analysed by recurrence quantification analysis (RQA). This method describes recurrence structures, including structures hidden within the series. The most important elements in recurrence plots are diagonal, vertical and horizontal lines as well as empty spaces. These elements describe typical features of a system, such as periodic behaviour, predictability, chaos-order and abrupt changes. The most popular recurrence quantifications include: recurrence rate (RR), determinism (DET), averaged and longest diagonal and vertical lines (L, L_MAX_ and V_MAX_), Shannon entropy and recurrence period density entropy (ENT, RPDE), laminarity (LAM), trapping time (TT), recurrence times (T1, T2), clustering coefficient (CC), and transitivity (TRANS). Definitions of all the above-mentioned recurrence quantifications are given in [42,50,51,52]. To derive more robust recurrence quantifications, we employed the sliding window method covering the main diagonal line of the RP. This method uses two parameters: a sliding windows size (w_s_) and a windows step (w_i_). Given that the amplitude of two analysed signals might differ, the sliding method was performed for a constant recurrence rate (against constant threshold). Then, a corresponding threshold ε was calculated for the obtained RR level. For every w_s_ the fraction of recurrence points was calculated, and quantifications were obtained.

Figure 7 illustrates the evolution of recurrence quantifications over time. It can be observed that DET, L, LAM, TT, V_MAX_, Clust and Trans (red lines) somewhat differ from the no-defect results (blue line) in the vicinity of data points 500–1000. This range is a confirmation of the existence of the defect that was detected by ultrasonic methods.

The difference in the values of both signals results from the presence of a local defect, which means that the defect location can easily be established. In the remaining time periods the recurrence time evolution is similar for all quantifications. That means that the above-mentioned quantifications can be used as real defect indicators during the drilling of CFRP materials.

#### 3.2.2. Recurrence Quantification Analysis of GFRP Drilling Signals

The same recurrence analysis method as above was used for the GFRP material. Obtained drilling forces are shown in Figure 8. For this case the forces are over two times smaller, i.e., about 250 N. However, the no-defect (blue line) and defect (red line) forces have very similar patterns, and the defect itself is difficult to spot. Therefore, the recurrence analysis was applied again to detect any differences between both signals.

Recurrence plots are presented in Figure 9. These plots were obtained for the same embedding parameters (*m* = 5, *d* = 2) after time normalization. The RP pattern for the drilled signal without a defect is practically the same in the entire plot (Figure 9a). An analysis of the RP reveals the presence of many points and short line segments with different orientations. This region is also responsible for the defect location as in previous material (CFRP). In this case the depth of defect was equals about 1.6 mm.

Nevertheless, an analysis of the RP obtained for the drilled material with a defect (Figure 9b) shows the presence of a region with a small number of points. This region is located close to data points of 500–2500, which is related with the location of the defect detected by ultrasonic testing. This recurrence pattern is probably caused by the defect inside the GFRP material detected by C-scanning. In general, an in-depth inspection of recurrence plots makes it possible to select places (times) at which dynamic behaviour transitions occur.

Results of recurrence analysis by the sliding window technique are given in Figure 10. The recurrence indicators DET, L, ENT and TT are very important for defect detection because they show some differences between both signals. Variations in the values of these RQA parameters could show differences between both system states and could help locate the defect. It should be noted that two parameters: L and ENT, have higher average values for the signal with the defect in the total range of the analysed signals. It can be concluded that for the case of GFRP the average length L and ENT have higher values for the signal with a defect.

The RP method is a suitable tool for detecting defects based on the time series in the cutting process. Dynamical changes in the RP structure and RQA values can be used to identify and quantify transitions between different system states and hence to detect defects.

## 4. Conclusions

This paper presents the results of a study investigating real defect formation in composite materials. Two types of polymer composites (with carbon and glass fibres) were tested. The first stage of the study involved performing ultrasonic analysis to detect the presence of real defects. The use of the through-transmission method made it possible to locate a real defect inside the composite samples. The choice of the ultrasonic method depended on the size of the defects, access to the material and the limitations of other methods used in the research. For example, shearography testing is mainly used for delamination, magnetic testing is used for defects larger than 1 mm, and the visual methods are limited only to the external surface. Based on the analysis of the literature, it can be concluded that, in addition to ultrasonic testing, radiography testing using X-rays could also be used for composites testing. A defect with a size of 1.3 mm (CFRP) and 1.1 mm (GFRP) was detected by ultrasound testing. After that, drilling was carried out, during which the force values were recorded. The forces were then used in nonlinear analyses. The nonlinear analyses consisted of creating recurrence plots to show location and depth of the real defect. The results of the study made it possible to create these plots with the exact determination of the size and location of the defect in the two tested polymer composites. Additionally, recurrence quantifications were performed as a complement to the recurrence method in order to determine the location and size of the real defect in the composite materials. The results have shown that DET, L and TT are the most suitable quantifications for detecting real (not artificial) defects in CFRP and GFRP.

## Figures and Tables

**Figure 1 materials-15-07335-f001:**
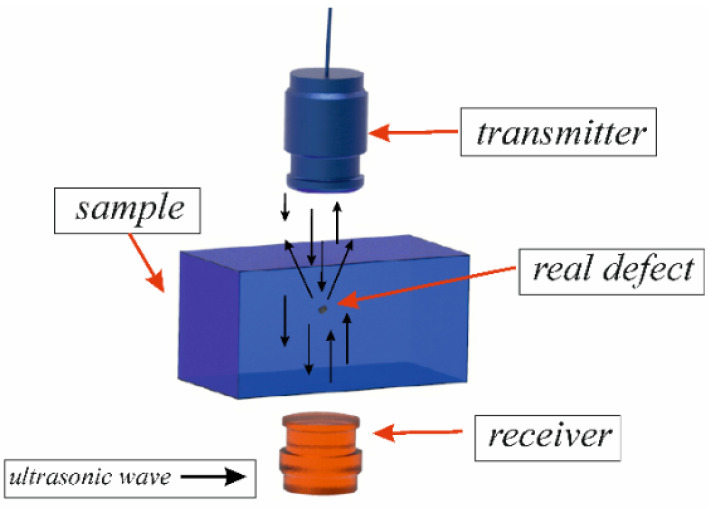
Schematic design of through-transmission method.

**Figure 2 materials-15-07335-f002:**
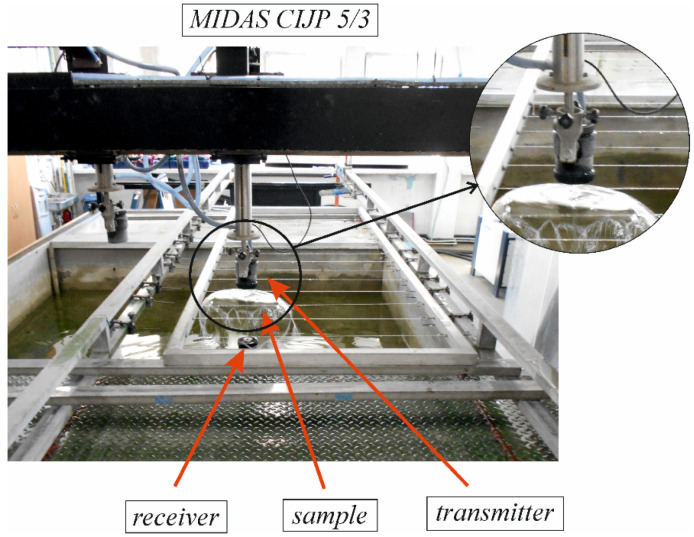
Experimental setup for tests by through-transmission method.

**Figure 3 materials-15-07335-f003:**
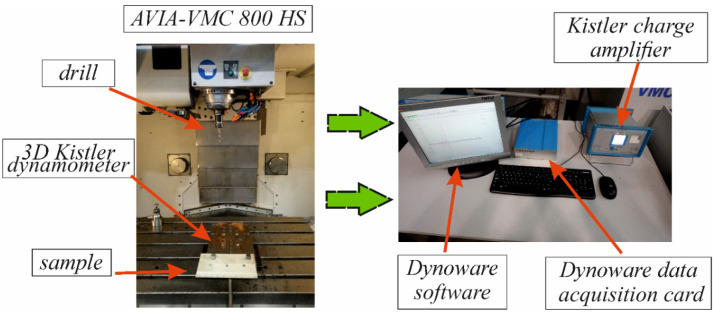
Setup of the machining process.

**Figure 4 materials-15-07335-f004:**
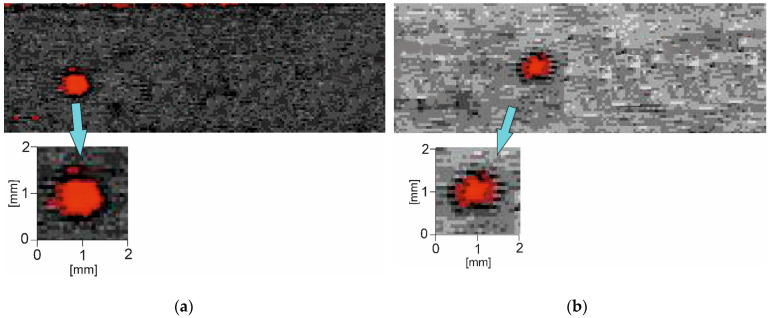
C-scan images obtained by through-transmission method for (**a**) CFRP and (**b**) GFRP.

**Figure 5 materials-15-07335-f005:**
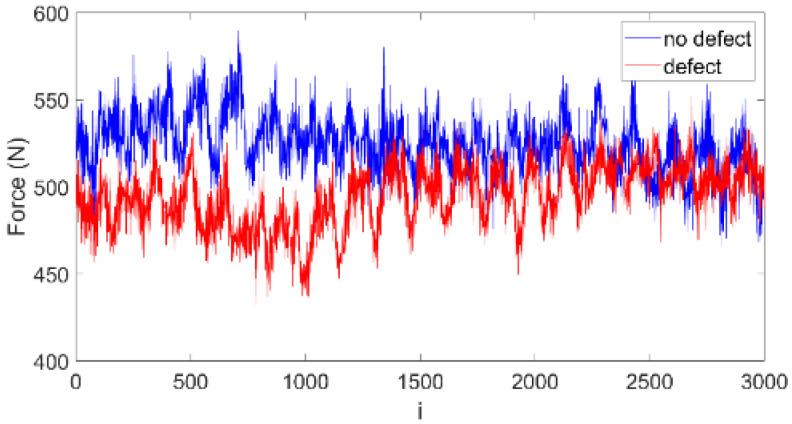
Cutting forces measured when drilling CFRP without defect (blue line) and with defect (red line).

**Figure 6 materials-15-07335-f006:**
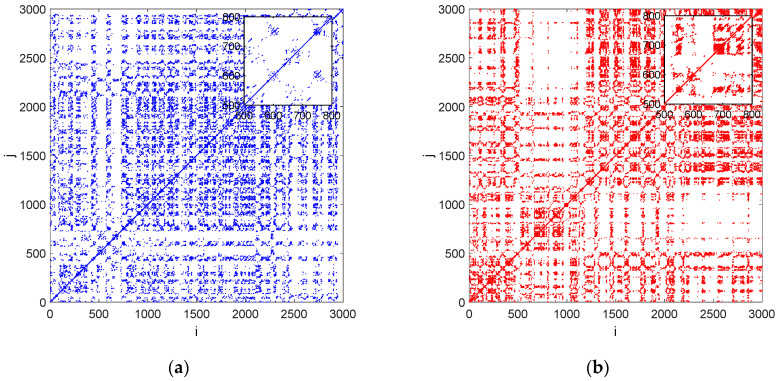
Recurrence plots showing the cutting force in drilling CFRP without defect (**a**) and with defect (**b**). The calculations were made for constant ε = 0.5.

**Figure 7 materials-15-07335-f007:**
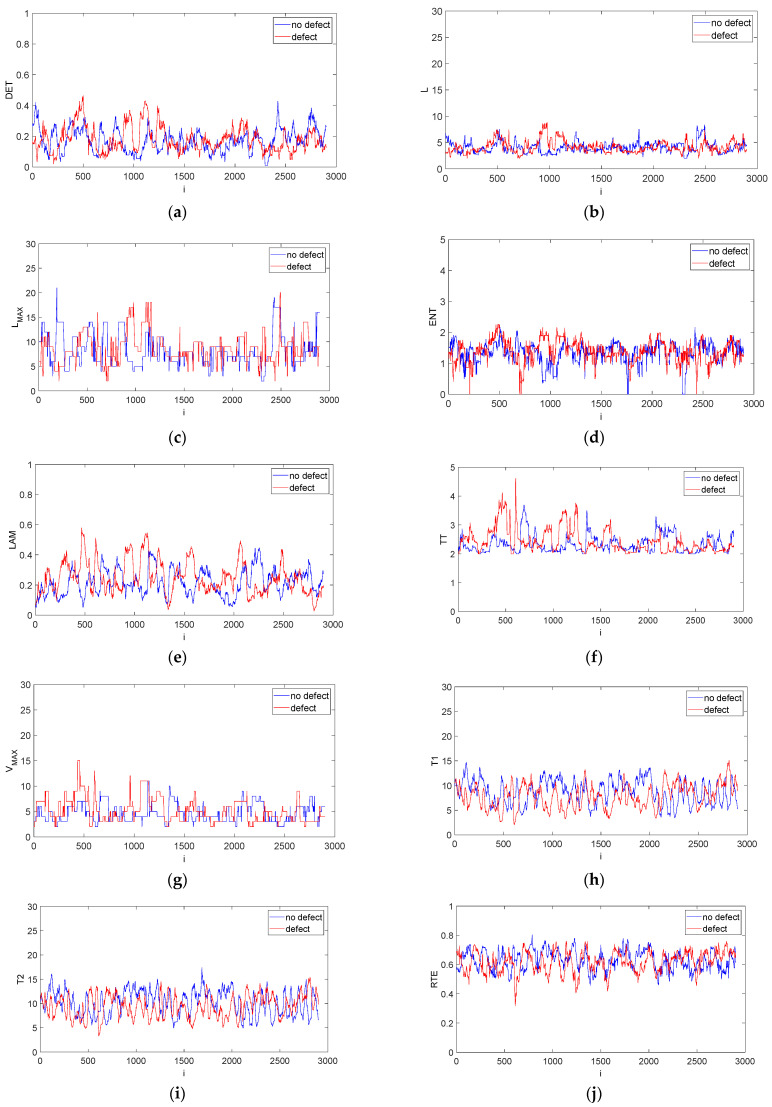
Recurrence quantification time evolution obtained by moving window technique for CFRP without defect (blue line) and with defect (red line): DET (**a**), L (**b**), L_MAX_ (**c**), ENT (**d**), LAM (**e**), TT (**f**), V_MAX_ (**g**), T1 (**h**), T2 (**i**), RTE (**j**), CC (**k**) and Trans (**l**). The calculations were made with w_s_ = 100 and w_i_ = 1. The quantifications show system behaviour over time.

**Figure 8 materials-15-07335-f008:**
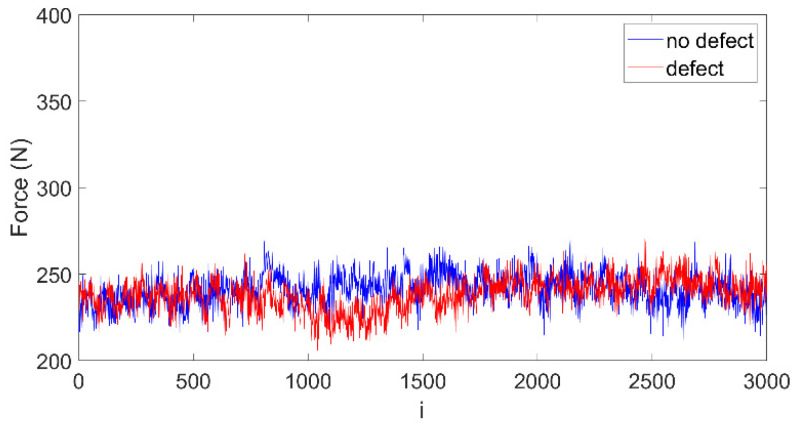
Cutting forces measured during drilling of GFRP without defect (blue line) and with defect (red line).

**Figure 9 materials-15-07335-f009:**
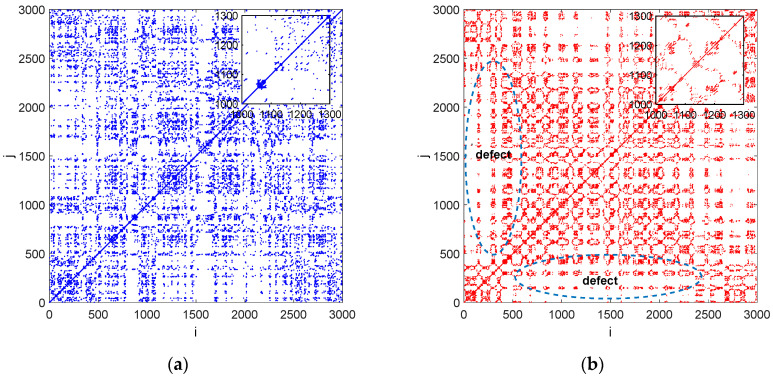
Recurrence plots showing the cutting force in drilling GFRP without defect (**a**) and with defect (**b**). The calculations were made for constant ε = 0.5.

**Figure 10 materials-15-07335-f010:**
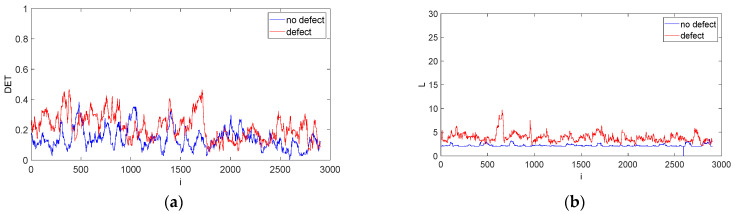
Recurrence quantification time evolution obtained by moving window technique for GFRP without defect (blue line) and with defect (red line): DET (**a**), L (**b**), L_MAX_ (**c**), ENT (**d**), LAM (**e**), TT (**f**), V_MAX_ (**g**), T1 (**h**), T2 (**i**), RTE (**j**), CC (**k**) and TRANS (**l**). The calculations were made with w_s_ = 100 and w_i_ = 1. The quantifications show system behaviour over time.

## Data Availability

Not applicable.

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
