# Peer review of "Non-Destructive Detection of Real Defects in Polymer Composites by Ultrasonic Testing and Recurrence Analysis"

_materials, 2022, doi:10.3390/ma15207335_

Round 1

Reviewer 1 Report

Non-destructive Evaluation (NDE) is an effective quality technology for polymer matrix composites (PMC) and related structures, enabling the detection of internal defects and imperfections in material structure. Two typical NDE methods, ultrasonic testing and recurrence analysis, were employed in this investigation to detect real defects inside the carbon fiber reinforced plastics (CFRPs) and glass fiber reinforced plastics (GFRPs) composite materials. Based on the experimental results and analytical outcomes, adaptation of these two methods in detecting the defects formed via different routes has been presented by the authors. This submission is carefully prepared and would be beneficial to enhancing application of NDE technology in fabrication of PMC structures. To meet the strict publication requirements of MDPI, some improvement of this submission is still necessary, and the followed comments is suggested to be considered as modifying the manuscript.

(1). The NDE analysis methods applied in this study is adapted for the PMCs with different matrices such as PI or BMI in addition to EP?

(2) The influence of defect size, location and characteristics on the detection accuracy? And also the volume fraction of the reinforcement fiber.

(3) The conclusions of this work was obtained from experiments carried out with plate samples, and is it same for the complicated shape samples, such as surface sample?

Author Response

Responses to comments from Reviewer 1
We would like to thank the Editor for their consideration, and the Reviewers for the time spent on carefully reviewing this work and for their valuable deep insight and comments. We feel that this paper is now clearer, more thoroughly discussed and better-referenced. 
The work has been revised to address the reviewers’ suggestions. In attachmen find hereafter a point-by-point reply to the comments and suggestions. Any revisions to the manuscript was marked up using the “Track Changes”.

Yours sincerely,

Krzysztof CiecielÄ…g
Krzysztof Kęcik
Agnieszka Skoczylas
Jakub Matuszak
Izabela Korzec
Radosław Zaleski

Reviewer 2 Report

In the manuscript entitled "Non-destructive detection of real defects in polymer composites by ultrasonic testing and recurrence analysis", C. CiecielÄ…g et al. presents results of ultrasonic non-destructive testing of carbon fibre-reinforced plastics (CFRPs) and glass-fibre reinforced plastics (GFRPs).

In my opinion, none novelty is reported in the present manuscript; in what is the present study different from those already published? Discuss.

Add more references in the manuscript.

In the abstract, the authors should eliminate the first sentence, and start from "This paper ...".

In the introduction, the authors should compare the destructive testing with the non-destructive ones; moreover the authors should briefly describe all used non-destructive testing methods, e.g. X-ray CT, reporting for each of them the advantages and disadvantages. The obtained results should be compared with other standard techniques to evaluate the method accuracy; indeed, the authors should analyse a reference material to evaluate the effectiveness of reported results.

Which is the measurement reproducibility? Discuss.

I can accept this manuscript with major revision.

Author Response

Responses to comments from Reviewer 2
We would like to thank the Editor for their consideration, and the Reviewers for the time spent on carefully reviewing this work and for their valuable deep insight and comments. We feel that this paper is now clearer, more thoroughly discussed and better-referenced. 
The work has been revised to address the reviewers’ suggestions. In attachmen find hereafter a point-by-point reply to the comments and suggestions. Any revisions to the manuscript was marked up using the “Track Changes”.

Yours sincerely,

Krzysztof CiecielÄ…g
Krzysztof Kęcik
Agnieszka Skoczylas
Jakub Matuszak
Izabela Korzec
Radosław Zaleski

Reviewer 3 Report

Overall, the paper presents a good case study on non-destructive detection of defects in polymer composites using ultrasonic testing method. There are no major concerns on the study, but a few suggestions for the authors. 

1, The paper has a long wind introduction to cover a review, theory and aim for the study. It is not clear to me. It is suggested to short the section and put it into two parts. 

2, Experiment setup is not very clear to me. Is the defeat created by drilling? What can the measurement resolution be achieved? Description of method needs to improve. 

3, The authors presented plenty of waveforms obtained, but it is hard to understand these waveforms and correlations to the defects. A better discussion is needed. 

4. Some English mistakes can be found such as Page 1, line 28-29. Some places need a proper reference such as Page 2, 2nd paragraph. 

Author Response

Responses to comments from Reviewer 3
We would like to thank the Editor for their consideration, and the Reviewers for the time spent on carefully reviewing this work and for their valuable deep insight and comments. We feel that this paper is now clearer, more thoroughly discussed and better-referenced. 
The work has been revised to address the reviewers’ suggestions. In attachmen find hereafter a point-by-point reply to the comments and suggestions. Any revisions to the manuscript was marked up using the “Track Changes”.

Yours sincerely,

Krzysztof CiecielÄ…g
Krzysztof Kęcik
Agnieszka Skoczylas
Jakub Matuszak
Izabela Korzec
Radosław Zaleski

Reviewer 4 Report

The manuscript "Non-destructive detection of real defects in polymer composites by ultrasonic testing and recurrence analysis" has an actual subject of the applied research on materials.

The study perform a non-linear analysis of the drilling process for glass and carbon fibre-reinforced plastics with real defects formed in the composite materials at the manufacturing stage.

The methods and procedures were adequate, the presented data are interesting and the conclusions was based on experimental results.

However, the introduction part is too large and some valuable and actual references were lost!

Also, the scientific part of contribution must be emphasized.

.

Author Response

Responses to comments from Reviewer 4
We would like to thank the Editor for their consideration, and the Reviewers for the time spent on carefully reviewing this work and for their valuable deep insight and comments. We feel that this paper is now clearer, more thoroughly discussed and better-referenced. 
The work has been revised to address the reviewers’ suggestions. In attachmen find hereafter a point-by-point reply to the comments and suggestions. Any revisions to the manuscript was marked up using the “Track Changes”.

Yours sincerely,

Krzysztof CiecielÄ…g
Krzysztof Kęcik
Agnieszka Skoczylas
Jakub Matuszak
Izabela Korzec
Radosław Zaleski

Round 2

Reviewer 2 Report

Now, in my opinion, the revised version of the manuscript can be accepted for publication.